# Demographic isolation and attitudes toward group work in student-selected lab groups

**Mitra Asgari** [ID][1☯]*, **Amy E. Cardace**[2☯], **Mark A. Sarvary** [ID][3]

**1** Division of Biological Sciences, University of Missouri, Columbia, Missouri, United States of America,
**2** School of Education, Fairleigh Dickinson University, Teaneck, New Jersey, United States of America,
**3** Investigative Biology Teaching Laboratories, Department of Neurobiology and Behavior, Cornell University, Ithaca, New York, United States of America

☯ These authors contributed equally to this work.
* mitra.asgari@missouri.edu

## Abstract

Small group work has been shown to improve students' achievement, learning, engagement, and attitudes toward science. Previous studies that focused on different methods of group formation and their possible impacts mainly focused on measures of students' academic ability, such as GPA, SAT scores, and previous familiarity with course content. However little attention has been given to other characteristics such as students' social demographic identities in research about group formation and students' experiences. Here, we studied the criteria students use to form lab groups, examined how the degree of demographic isolation varies between student-selected and randomly-formed groups, and tested whether demographic isolation is associated with group work attitudes. We used a pre-post survey research design to examine students' responses in a large-enrollment biology laboratory course. Descriptive analyses showed that "students sitting next to me" (57%) followed by the combination of "students sitting next to me" and "friends" (22%) were the two most common criteria students reported that they considered when forming research groups. Notably, over 80 percent of students reported forming groups with those who sat nearby. We studied instances where students were isolated by being the only members of a historically marginalized population in their lab groups. The prevalence of demographic isolation in student-selected groups was found to be lower than in the simulated randomly assigned groups. We also used multilevel linear regression to examine whether being an isolated student was associated with attitudes about group work, yielding no consistent statistically significant effects. This study contributes to growing knowledge about the relationship between students' demographic isolation in groups and group work attitudes.

**Data Availability Statement:** This study involves human research participant data. All relevant data that were used in analyses are first de-identified and then provided as Supporting Information files to allow to replication of the results of the study.

## Introduction

How the demographic composition of small groups influences students' experiences has received increasing attention in recent years. Previous research has addressed group dynamics, students' affective outcomes (i.e., attitude, feeling), or learning in college classrooms [e.g., 1–

**Funding:** The author(s) received no specific funding for this work.

**Competing interests:** The authors have declared that no competing interests exist.

3]. Compared to competitive learning environments, working in collaborative learning settings could improve the opportunities for participation of historically marginalized groups in STEM, such as students who self-identify as female, African American, and Hispanic, among other identities [4–7]. However, poor group dynamics may lead to academic intimidation, where students may feel less competent [8], more anxious, quieter during group discussions [9], or underperform in the presence of their peers [10] due to phenomenon such as stereotype threat [11].

Cooperative learning, also known as small group work [12, 13], is philosophically rooted in social interdependence theory [14] and has become one of the most used and well-studied student-centered instructional practices [15]. In cooperative learning, students often work in small groups focusing on a set of shared learning goals and being assessed by the instructor both at the individual and the group level [16]. By incorporating small group work in classrooms, instructors can provide students with opportunities to discuss their ideas and perspectives with each other, provide and receive peer feedback, and develop and improve skills such as scientific reasoning, argumentation, communication, and teamwork [17, 18]. The value of group work for students' learning and attitudes toward science has been studied and promoted for years [19–21]. Meta-analyses of group work studies have shown improvement in students' learning, interest in the subject, self-esteem, acceptance of diversity [22], academic achievement, persistence in coursework, and attitude toward learning [23]. More recent research has also shown enhancement of students' overall achievement and learning [24–26], engagement [27], use of high-order cognitive skills, attitudes toward science and persistence in STEM courses related to using group work [16, 26, 28, 29].

When permanent or long-term small group work is incorporated in classrooms, sometimes referred to as formal group work [16, 30], often one of the following group formation strategies is used. Groups can be formed by students, sometimes referred to as self-selected, student-selected, or student-formed groups, with little or no interference from an instructor. Alternatively, students can randomly be assigned to groups, where no criteria other than final group size are used [23]. Finally, groups could be created by using one or multiple criteria, often referred to as instructor-formed or instructor-assigned groups [31]. Based on existing literature in STEM education, the most common criteria instructors use when forming groups are measures of students' academic ability, such as GPA, SAT scores, prior related courses, previous familiarity with course content [32–36] or learning styles [37–39]. However, little attention has been given to other students' characteristics such as their social demographic identities and background when thinking about group formation.

Previous studies that have investigated group work quality, dynamics, and students' learning in relationship to the demographic composition of the groups mainly focused on students' gender identity. When working in small groups in STEM college classrooms, female students expressed less comfort than male students [8]. In an introductory psychology course, in groups of three, female students were less task-oriented in mixed-gender groups than in same-gender groups and they were less talkative when they were solo in groups compared to male students [10]. In another study, female engineering students showed more anxiety in female-minority groups in their first year of college. Findings of the same study indicated less verbal participation of female students in female-minority groups regardless of their academic year in college compared to the sex-parity groups and female majority groups [40]. Student collaboration, observed as equitable group work processes, was stronger in gender-balanced groups compared to all-male or solo-male groups [41], although no difference was observed in students' performance [10, 41]. In another study, researchers found that grouping by gender mostly impacts students' attitudes toward instruction rather than their performance and that gender-balanced and female-only groups are the most effective [42].

These findings suggest the possibility of similar experiences by other social minority groups when it comes to group work. A study of small groups in STEM college classrooms showed that students identified as *Underrepresented Minority (URM) students* reported higher levels of social-comparison concern than the majority students [8]. During peer discussions in a large introductory biology course, underserved American and Asian-American students showed a stronger preference for the role of listener over leader/explainer when compared to white Americans [9]. In an introductory sociology course, when the associations between leadership, sex, race, and course performance were investigated in teams created by the instructor, researchers found that white students had more leadership roles in teams, received higher grades, and were evaluated higher than students of color [43]. The research findings related to students from other social minority groups such as first-generation college students or international students are even more scarce. During peer discussions, international students also showed a stronger preference for having the role of listener over the leader/explainer role when compared to white Americans, and they reported experiencing higher anxiety during peer discussions [9].

In addition to the above findings, a body of research shows that when given the opportunity, students in science classrooms tend to create more homogenous groups based on gender and ethnicity [29] and based on a combination of gender identity, academic, and personality characteristics [44]. We were eager to explore similar questions in a smaller learning setting, the laboratory part of a large-enrollment science course. This study was conducted to understand the students' considerations when forming groups, the frequency of demographic isolation in student-formed groups, and the relationships between group compositions and students' attitudes about group work. We used a pre-post survey approach and included questions to learn about students' demographic identities, criteria considered when forming their groups, and group work attitudes. We assessed the frequency of students' demographic isolation in self-selected groups compared to hypothetical randomly-formed groups. We use the term *demographic isolation* to describe the situation where a student is the only member from a particular demographic group in their lab group (e.g., the only female or ethnic minority student in a group). We also investigated whether this isolation is associated with differences in students' attitudinal group work scores. Our research questions are listed below:

Q1. What criteria do students report that they consider when forming lab groups?

Q2. How does the composition of student-selected groups differ in terms of demographic isolation when compared to hypothetical randomly-formed groups?

Q3. Do students' attitudes toward group work vary between students who were demographically isolated in groups and those who were not?

## Materials and methods

### Study context

**Course setting/Information.** We conducted our study in an introductory biology course at a large research university in the northeastern United States. This two-credit-hour inquiry-based laboratory course is required across many biology-focused majors. Approximately 400 first- and second-year students enroll in the course each semester. The course consists of a weekly 50-minute lecture [45, 46] and a 3-hour laboratory (hereafter lab) session [47], both using evidence-based teaching practices. In this course, the students could enroll in any of the twelve lab sections offered each semester. Group formation and group work take place in the lab sessions, which were held in multiple small rooms with traditional fixed tables and movable

seating, each hosting a maximum of 18 students. Labs were led by graduate teaching assistants (hereafter GTA) and offered guided inquiry learning environment throughout the semester [48, 49]. In both lecture and lab environments, students were free to self-select both their seats and their lab groups.

**Group formation and group work in the lab.**   For the first three weeks of the semester, students worked together informally during labs on activities that were graded and assessed at an individual level. During the fourth week in the lab, GTAs asked students to form groups of three (hereafter student-selected groups). Students usually stayed in these groups for the rest of the semester. In cases where the number of students per lab was not perfectly divisible by three, a few groups of 2 were allowed to be formed. For the rest of the semester, students were expected to work with their group members on various activities such as designing and conducting an experiment, collecting and analyzing data, and group presentations. After group formation in week 4, students were assessed and graded both at the individual and group levels for the rest of the semester.

## Study design and data collection

We used a survey research approach to quantitatively address three questions about (1) the criteria students considered when forming their lab groups, (2) the prevalence of students being demographically isolated in their self-selected groups compared to randomly-formed groups, and (3) relationships between being demographically isolated and attitudinal outcome measures. The course and survey administration timelines are summarized in Fig 1, specifying when groups were formed and the related data were collected. To account for possible variation between semesters, we studied these questions for two consecutive semesters, Spring and Fall 2019. The pre-and post-surveys were conducted online via Qualtrics (Seattle, WA). In the pre-survey, participants responded to a combination of questions asking about their demographics, academic backgrounds, and attitudes toward group work. In the post-survey, in addition to the same pre-survey questions, students were asked to share the criteria they used to form a formal lab group in week 4.

## Ethics statement

This project was approved by the Cornell University Institutional Review Board and has been granted an exemption from IRB review (#1901008516).

## Participant recruitment

Participants were students who enrolled in the course. To recruit participants, we used the course learning management system (LMS) at the beginning and end of each semester to share a message providing the project summary information, with a link to the survey and a written consent form with students. A few bonus course credits for responding to each of the pre-and

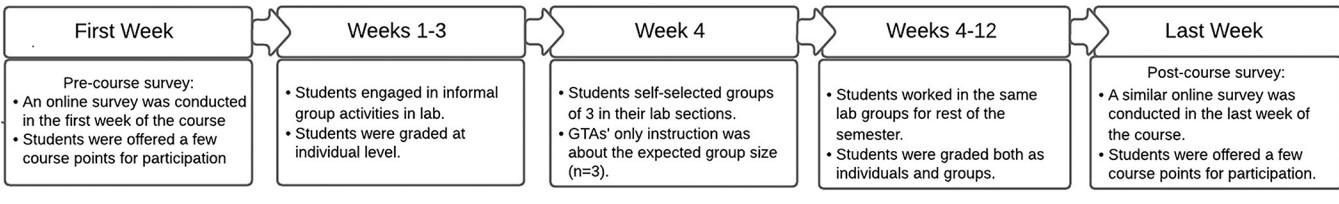

**Fig 1.  The timeline and processes of group formation and data collection.**

post-surveys were also provided to students. Responses from students who did not provide consent or were minors were removed from the study before the analyses.

## Survey measures

**Students' demographic information.**   The demographics we examined in this study were students' gender and ethnic-racial identities, college generation status, and international student status. We focused on the first two demographics due to the vast evidence about the historical marginalization of individuals who identify as female, Black/African American, Hispanic/Latinx, and Native American/American Indian/Indigenous American in STEM higher education [50]. Previous research also shows that first-generation college students, individuals whose parents do not have a 4-year bachelor's degree, face various academic challenges in STEM courses [51–53]. We also focused on international students because they are often in the minority in US undergraduate college classrooms, and there is little published research about this population in the groupwork STEM education literature [see 9]. In this context, international students are referred to citizens of other countries who came to the United States for post-secondary education [54]. These data helped us assess the demographic composition of student-selected groups and were used as independent variables in multi-level regression analyses predicting groupwork attitudes.

**Criteria students considered when forming lab groups.**   In addition to demographic information, a multiple-choice question was included in the post-survey each semester to learn about the criteria students considered when forming their group during the fourth week of the lab. Students could select more than one option from a list of twelve responses including the following: "friends"; "students sitting next to me in the lab"; "students' GPA and familiarity with the biological concepts"; "students' gender identity"; "students' race/ethnicity"; "students' year in college." This survey question with full list of response choices provided to students, can be found in the supporting information document. In terms of the logic behind the answer choices selected, when the survey questions were being developed the authors considered the most probable choices such as "friends", "proximity in the lab" and explored the STEM education literature for the most common visible and hidden identities and characteristics students may identify with and commonly studied. We thought of identities that could be visible from the first day of the class when students meet each other like "racial/ethnic identity" and the ones that may be less evident or hidden but students could possibly share with each other in the first 3 weeks of the lab and before forming groups, such as "being an international student or not". In the post-survey before students see this multiple-answer question, they were also asked to respond to an open-ended question of "Which criteria did you use to form your research group?". The preliminary coding of students' responses for the open-ended version of this question yielded similar findings and no additional themes. Thus, to avoid redundancy, here we only present outcome of the multiple-choice question.

**Student attitudes about group work.**   We adapted and re-validated an item set from the Student Attitudes towards Group Environments (SAGE) survey [55] to assess students' group work attitudes in this context. The original questionnaire had 54 five-point Likert survey items (strongly agree, agree, undecided, disagree, strongly disagree) categorized by four factors: quality of product and process, peer support, student interdependence, and frustration with group members.

Our process of evaluating the original SAGE items and selecting a subset for current use leveraged strategies for improving validity during instrument design [56]. We mapped the factors of interest to particular items and engaged content and course experts (M.A and A. E. C) to check the items for student accessibility. Considering previously reported relationships

between demographic group composition and student group processes or performance noted above [10, 41, 43], we chose to focus on two particular factors, work quality and interdependence. Researchers affiliated with the course screened the items to make sure they were logically related to the students' lab group activities (M.A and M. A. S).

This process yielded fourteen items that were most relevant to our study (see S1 Table). We analyzed data from all respondents who replied to all 14 items, using a generalized Rasch model for polytomous data [57]. We found that the14-item instrument showed strong internal consistency statistics of 0.79 for the pre-test (n = 547) and 0.84 for the post-test (n = 281). In addition to establishing content validity as noted above, we also checked estimates from the model for expected patterns; all weighted mean square fit statistics were within the expected range (0.77–1.33) and the response category thresholds within each item were ordered from low to high as theorized [57].

## Participant demographics

The data set included 1148 responses, including pre-and post-surveys for both semesters. The response rate of students after data cleaning was 74% and 80% for the spring and fall semesters, respectively. The participation rates and the demographic breakdowns of participants were generally consistent between the two semesters (Table 1). In both semesters, females outnumbered males by almost two to one. The responses of students who selected non-binary or prefer not to say options had to be eliminated from the quantitative analysis due to their small sample size (0.7% and 0.6% in Spring and Fall 2019). White and Asian students comprised the majority of students in both semesters. Due to the relatively small number of students who self-identified with historically marginalized races and ethnicities, we pooled their responses as a single category for the quantitative analyses. This group included students identifying as (1) Black/African American, (2) Hispanic/Latinx, (3) Native American/American Indian, (4) Native

**Table 1. Response rate and demographic overview of students who participated in the study over the two semesters.**

|  | Spring 2019 | Fall 2019 |
| --- | --- | --- |
| **Number of students enrolled in the course** | 378 | 392 |
| **Number of respondents*** | 281 | 315 |
| **Response rate** | 74% | 80% |
| **Gender** |  |  |
| Female | 64% | 65% |
| Male | 35% | 34% |
| Nonbinary | 0.7% | 0.6% |
| Prefer not to say | 0.3% | 0.3% |
| **Race/Ethnicity** |  |  |
| White | 37% | 40% |
| Asian | 32% | 34% |
| Hispanic, Latinx, or Spanish origin | 13% | 9% |
| Black or African American | 11% | 11% |
| American Indian or Alaska Native | 0% | 0% |
| Native Hawaiian or Other Pacific Islander | 0% | 1% |
| Middle Eastern or North African | 2% | 1% |
| Other Race, Ethnicity, or Origin | 5% | 3% |
| **First-generation college status** | 17% | 16% |
| **International status** | 4% | 7% |

Hawaiian or Other Pacific Islander, (5) Middle Eastern or North African, or (6) Some Other Race, Ethnicity, or Origin. We represent this combined group with an acronym to identify the primary groups, BHN+. As discussed by previous studies [cf. 9, 58–60], student populations from these backgrounds share historical marginalization and present underrepresentation in American post-secondary STEM environments [61]. Although we acknowledge these sub-groups may have varying experiences, exploring those differences is outside the scope of this study. The first-generation college students and international students constituted less than 20% and less than 10% of the participants, respectively (Table 1).

## Statistical analyses

We conducted three types of data analyses to address the research questions. All data used are shared as supporting information (S2–S4 Files). First, descriptive analyses were used to explore the criteria that students reported they considered when forming their lab groups and to analyze the demographic composition of student-selected groups of three. During this descriptive phase, we checked for the frequency of isolation of female students, BHN+ students, first-generation students, and international students. Second, to examine whether student-selected groups yielded the same degree of student isolation as randomly assigned groups, we simulated random groupings by randomly assigning students to hypothetical groups of three within their lab sections with 100 iterations, yielding 100 possible versions of the random group assignments. We calculated the percentage of isolated students in each iteration and created a mean percentage for all 100 iterations for each demographic variable (female, BHN+, first-generation, and international students). We also compared the sample of 100 simulated percentages of isolated students to the actual percentage from student-selected groups and tested for a statistically significant difference using two-tailed one-sample t-tests.

Finally, we used three-level linear regression models (sometimes referred to as HLM or MLM) to examine whether students' demographic isolation status influenced their group work attitude scores. Given the clustered nature of our data set, we used random variables to more accurately account for variation due to the students' assigned GTAs, and lab group [62, 63]. In addition to student demographics, we included binary variables showing whether students were isolated in their lab group by gender, BHN+, first-generation, or international status.

Given that this study required both the estimation of student scores and the use of those scores as the dependent variable in linear regression, we used item response theory (hereafter IRT) and specifically a partial-credit model to estimate item difficulties and student scores [57]. We analyzed two types of student scores, Expected A Posteriori (EAP) scores that average multiple estimates, and plausible values (PV) that include a wider range of possible student scores to account for error more accurately. More details about these statistics can be found in the supporting information document (S1 File).

Given the continuous nature of the EAP and PV scores, we used a three-level linear regression with the *xtmixed* command in Stata15 [64], where each student (i) has a particular group (j) under a particular GTA (k):

Attitudes-post score$_{ijk}$ = $\beta_0$ + $\beta_1$(Attitudes-pre score$_{ijk}$) + $\beta_2$(Isolated-Female$_{ijk}$) + $\beta_3$(Isolated-First-generation$_{ijk}$) + $\beta_4$(Isolated-International$_{ijk}$) + $\beta_5$(Isolated-BHN+$_{ijk}$) + $\beta_6$(Female$_{ijk}$) + $\beta_7$(First-generation$_{ijk}$) + $\beta_8$(International$_{ijk}$) + $\beta_9$(BHN+$_{ijk}$) + $B_{10}$(Semester$_{ijk}$) + $u_1$(GTA)$_{jk}$ + $u_2$(group)$_{jk}$ + $\varepsilon_{ijk}$

○ Attitudes-post score is group work attitudes post plausible value score

○ Attitudes-pre score is group work attitudes overall pre plausible value score

○ Isolated-Female, Isolated-first-generation, Isolated-International, and Isolated-BHN+, are four binary variables reflecting if a student with these demographics were isolated in their lab group (0 = not isolated, 1 = isolated)

○ Female, First-generation, International, and BHN+ are four binary variables describing whether a student was from any of these demographics (0 = no, 1 = yes)

○ Semester is the semester a student took the course (1 = spring, 2 = fall)

○ GTA identifies the GTA-led lab section, and the group identifies the lab group

NOTE: The estimated random coefficients for GTA and lab groups are represented by $u_1$ and $u_2$, respectively. The estimated error is represented by $\varepsilon$.

## Results

### Question 1: What criteria do students report that they consider when forming lab groups?

In response to the post-course survey question about the criteria considered when forming their lab groups, a majority of students reported forming groups with peers sitting next to them in the lab (59% in Spring 2019 when n = 236; 54% in Fall 2019 when n = 208). The next common criteria selected were peers sitting next them plus friends (27% in Spring2019 and 16% in Fall 2019), followed by people sitting around them, friends, and other criteria (11% in Spring 2019 and 25% in Fall 2019) and only friends (3% in Spring19 and 5% in Fall 2019) (Fig 2). In the Fall 2019 post-course survey a few additional questions were asked to better understand the stability of lab groups and seating choices. Most students (93%) reported that they stayed in their original lab groups for the rest of the semester and did not change their groups. Students also stated that on the first day of class, they chose the first available lab seat (74%) or

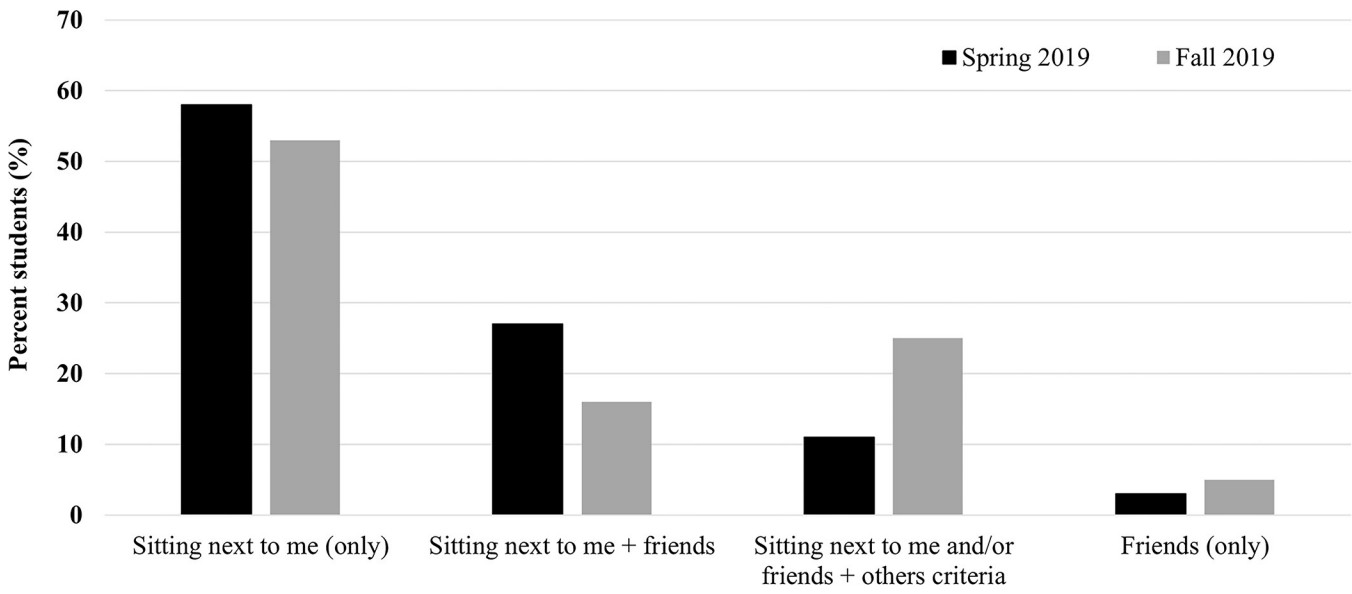

**Criteria students used when forming groups**

**Fig 2. The criteria most students reported considering when forming groups.** The "sitting next to me (only)" or "Friends (only)" represents the students who only selected one of these two items as criteria to form groups. Students' selection of more than one criterion is shown with a + sign.

sat by their friends (21%) in response to the questions on how they decided where to sit in the lab. Most students (87%) also reported that they did not change where they sat in the lab between the first week of class and when forming their groups in week four.

### Questions 2. How does the composition of student-selected groups differ in terms of demographic isolation when compared to hypothetical randomly-formed groups?

If students' processes for selecting groups were not influenced by demographics, then we would expect to see similar demographic compositions for both self-selected and randomized groupings. To examine these compositions, we first analyzed the responses of students for whom data was available for all students in their group. This subsample included 303 students in 101 groups of 3 over the two semesters. The analysis of these data showed that 18% of these groups had isolated female students, 29% of groups had isolated BHN+ students, 31% of groups had isolated first-generation students, and 12% of groups had isolated international students. In terms of the number of students rather than groups, 6% of female students, 10% BHN+ students, 10% of first-generation students, and 4% of international students were isolated in their lab groups.

Among the 303 students included in this analysis, the distribution of demographic categories generally mirrors that of the larger sample: female students comprised 62%; BHN+ students comprised 28%, first-generation students comprised 12%, and international students comprised 6%. Given that these demographic groups were not equally prevalent in the course, the social ramifications of being isolated would vary depending on the particular demographic. We examined this variation by calculating the percentage of students in each demographic group who were isolated. These adjusted percentages show a relative increase in the prevalence of isolation yielding 9% for female students, 33% for BHN+ students, 79% for first-generation college students, and 71% for international students.

Comparing the percentages of isolated students in hypothetical randomly assigned groups to the actual percentages presented in Table 2, we found more isolation in the randomly assigned groups among female and BHN+ students. The percentage of groups with an isolated female student rose from 18% in actual self-selected groups to 24% in simulated random groups. The percentage of groups with an isolated BHN+ student rose from 29% in the actual self-selected groups to 38% in the simulated random groups. In both cases, one-sample t-tests show statistically significant differences (females: t = 25.07, d.f. = 99, $p<0.001$; BHN+: t = 27.45, d.f. = 99, $p<0.001$). However, the percent of groups with isolated first-generation and international students did not show the same degree of variation between actual student-selected groups and simulated random groups (see Table 3). Examining the data for international students, there was a very small difference between the simulated mean percentage (12.8%) and the actual mean percentage (12.0%). This small difference would only generally affect one student in a class of one hundred students, yet it was statistically significant

**Table 2. Actual prevalence of demographic isolation in student-selected groups.**

| Students' Demographics | Actual percent of all groups with an isolated student | Actual percent of all students who experienced isolation | Actual percent of students in each demographic group who experienced isolation in their group |
|---|---|---|---|
| **Female** | 18% | 6% | 9% of female students |
| **BHN+** | 29% | 10% | 33% of BHN+ students |
| **First-generation** | 31% | 10% | 79% of first-generation students |
| **International** | 12% | 4% | 71% of international students |

**Table 3. Percent of students and groups with demographic isolation in simulated randomly assigned groups.** The values are grand-means for each demographic variable which are the average percentages of isolated students over 100 simulations.

| Students' Demographics | Simulated percent of groups with an isolated student* | Simulated percent of all students who were isolated* | Simulated percent of students in each demographic group who experienced isolation in their group |
|---|---|---|---|
| Female | 24% | 8% | 13% of female students |
| BHN+ | 38% | 14% | 44% of BHN+ students |
| First-generation | 31% | 10% | 82% of first-generation students |
| International | 13% | 5% | 76% of international students |

*Based on 100 randomly assigned groupings.

(t = 7.50, d.f. = 99, p<0.001). There was no statistically significant difference between the actual and simulated percentages for first-generation students (t = -1.71, d.f. = 99, p = 0.09).

At the individual student level, these differences between self-selected and hypothetical random groups yielded measurable differences in the percentages of female and BHN+ students who were demographically isolated shown in Fig 3. These differences were even higher when adjusting for the proportion of students from each demographic group in the course. (See the last columns in Tables 2 and 3).

### Question 3: Do students' attitudes toward group work vary between students who were demographically isolated in groups and those who were not?

We used two multi-level regression models to test for statistically significant relationships between demographic isolation and students' group work attitude scores (see S2 Table). Model

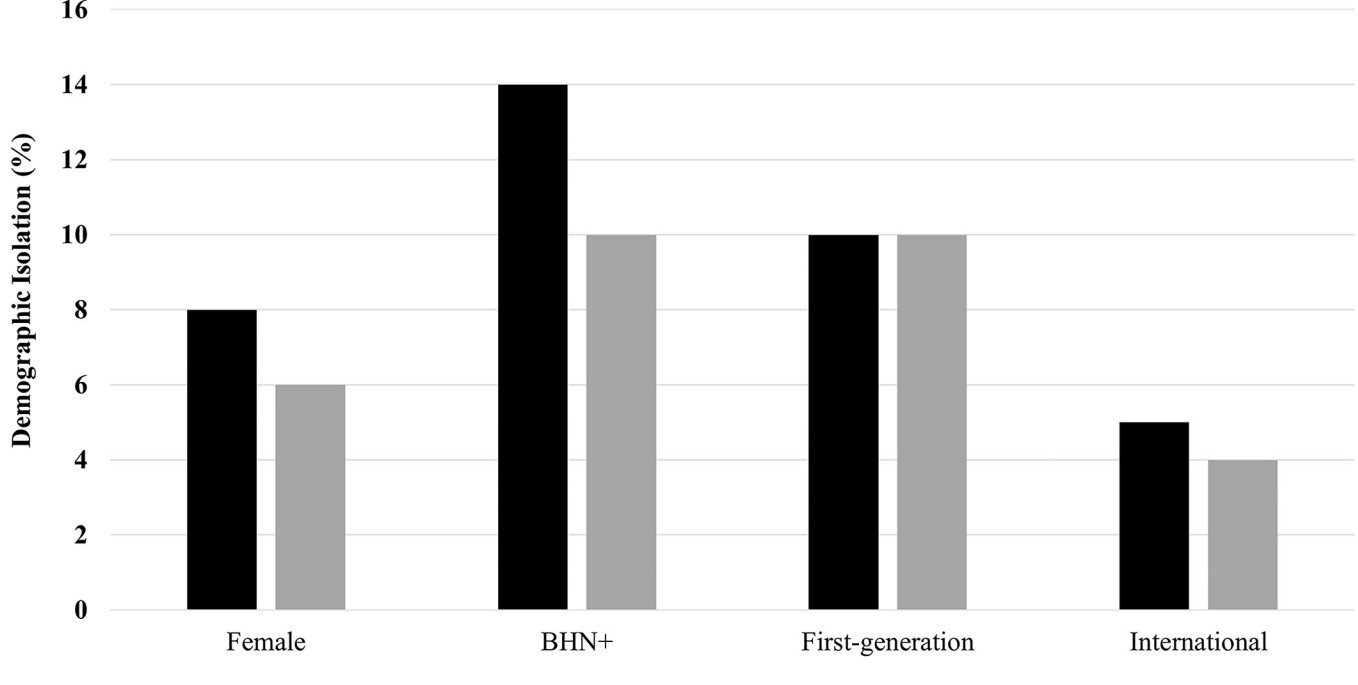

**Fig 3. Percent of students demographically isolated in actual self-selected versus simulated randomly assigned groups.**

**Table 4. Results from multi-level linear regression on posttest attitudinal scores controlling for pretest scores and estimating demographic and isolation variable coefficients (n = 185).**

| | | coeff. | s.e. | p-value | |
|---|---|---|---|---|---|
| **Fixed Effects** | Pre-Groupwork EAP | 0.689 | 0.072 | 0.000 | *** |
| | Female | -0.301 | 0.136 | 0.028 | * |
| | BHN+ | 0.115 | 0.180 | 0.523 | |
| | FGC | 0.290 | 0.520 | 0.577 | |
| | International | 1.004 | 0.463 | 0.030 | * |
| | Isolated Female | -0.057 | 0.257 | 0.824 | |
| | Isolated BHN+ | -0.023 | 0.256 | 0.927 | |
| | Isolated First-generation | -0.155 | 0.543 | 0.775 | |
| | Isolated International | -0.494 | 0.584 | 0.397 | |
| | Semester | -0.090 | 0.136 | 0.509 | |
| **Random Effects** | GTA | 0.000 | - | | |
| | Group | 0.286 | 0.131 | | |

* Denotes statistical significance at the 0.05 level, ** at the 0.01 level, and *** at the 0.001 level. LR test vs. linear model did not show a statistically significant difference: Prob > chi$^2$ = 0.5243

1 regressed the pretest attitudinal scores on the demographic and isolation variables and random variables that identified each student's teaching assistant and lab group. This model addressed whether students entered the course with group work attitudes that varied by isolation status. Model 2 regressed the posttest attitudinal scores on the pretest estimates and the other independent variables used in the first model. This model addressed the degree to which student scores showed pre-post change, and whether students exited the course with group work attitudes that varied by isolation status after controlling for their pretest scores.

Overall, the regression analyses showed no consistent negative associations between demographic isolation and students' groupwork attitudes, both as they began coursework and at the end of the course (Table 4). For the first model that used the pretest group work score outcome variable, the only statistically significant effect was a positive effect for students who were isolated as BHN+ students in their lab groups (see S2 Table). This effect was not consistent across the regressions we ran on plausible values, which better account for measurement error (see S3 Table). In the models we ran to examine associations with the posttest group work scores, there was no statistically significant effect of any demographic isolation variable. While the female and international demographic variables did show statistical significance using EAP scores, those effects were not consistent across the more accurate plausible value trials. The only independent variable that showed a consistent and statistically significant relationship with the group work post-test score was the pretest score (coeff. = 0.69; p<0.001).

## Discussion

Group work is a commonly used student-centered pedagogical approach in college classrooms. Given concerns about the social-emotional challenges some students face when isolated by particular identities, educators can benefit from a better understanding of student behaviors and related social experiences in these group work settings. In this examination of student survey data, we found that the majority of students reported casually forming groups with peers who were sitting around them and that their process yielded groups with less demographic isolation than would have resulted from randomly-formed groupings. Further, while demographic isolation has been identified in previous studies as presenting a variety of

challenges for students, our quantitative analyses did not show that demographic isolation was associated with variation in students' group work attitudes.

Students mainly reported choosing groups based on peers sitting close to them in the lab, selected as the sole factor or in addition to others such as friends. We acknowledge that current study cannot tease apart the conscious and unconscious biases that students' likely hold when locating themselves in group settings, interacting with new people, and navigating student-selected group formations. We also note that students may not be entirely forthcoming about their considerations. Yet, there was a consistent trend in students' responses that proximity and friendship were relevant for their choice of lab partners. Further, very few students reported that they intentionally considered demographics in their group choice. In the second semester post-survey, to better understand the consideration behind seating choices, when we asked students where in the lab, they decided to sit on the first day of class, most students (74%) mentioned choosing the first available seat they found or sat by their friends (21%); majority of them also reported not changing their seat between week 1 and when forming their lab groups in week 4. However, where students reported to sit could be related to the conscious and unconscious biases that students' likely hold which were beyond the scope of this study. Furthermore, we did not investigate the relationship between criteria students reported considering when forming groups and their attitude about group work. However, previous research that focused on informal group composition and group work, found that having a friend in a group was the main predictor of students' comfort in groups, a tendency reflected in our findings as well [65].

In our study, student-selected groups yielded another benefit for students, less demographic isolation when compared to hypothetical randomly formed groups. A previous study looking at the composition of informal groups in a large-enrollment biology classroom also showed students mostly formed homogenous groups in terms of ethnicity and gender [29]. This phenomenon is known as "homophily" [66] and is defined as the inclination of individuals to move toward and work with others who are like themselves. Our findings show this phenomenon even in lab settings where students are only forming groups from a small pool of peers (n ~ 15–18). Further, the prevalence of an identity group in the course related to the degree of isolation. Because the proportion of students who identified as first-generation or international was low in the course, there was more likelihood for those students to be isolated in lab groups, whether they self-selected or were randomly assigned. In contrast, females were consistently over-represented in this course, which led to fewer possibilities of being the only female in a lab group. The degree of isolation will always be dependent on the demographic composition of a particular context. Thus, we do not know if the statistical significance of gender and racial-ethnic demographics in this case will also be evident in other settings where the demographic composition of the cohort is meaningfully different. There must be a large enough number of students in a demographic category so that they have opportunities to change the composition of their group, and enough instances of that change to show measurable effects in a quantitative analysis.

Our findings indicate that allowing students to choose their groups can serve a self-protective purpose because they can avoid demographic isolation. While this study does not explain the mechanism by which self-selected groupings yield less isolation, the fact that there were statistically significant reductions in isolation when students self-selected warrants further inquiry. This is extremely important when we aim to support historically marginalized groups in STEM. Further, we note the difference between demographic traits that are visually evident to other students and those traits that are hidden. Henning et al. (2019) [67] noted the particular challenges that small-group coursework presents for students with particular political, religious, gender, and sexual identities. Given the hidden nature of some of these identities,

instructors are typically unable to systematically avoid isolating students by assigning groups based on identities that are visually evident or commonly tracked. Student-selected groups could allow students to choose peers they perceive to be allies in their groups even in the case of hidden identities by considering only those identities that seem important to them. This can be facilitated by providing time for students to get to know each other and form trust before forming groups, such as the first 3 weeks of informal group interaction in this study. In these conditions students can choose groups to avoid the types of demographic isolation that they perceive as negative. For that reason, we estimate that the impact of any measurable demographic isolation that remains after students form groups may be minimized.

Other studies show the benefit of student-selected groups compared to randomly-formed groups when it comes to team experience [68]; student satisfaction and grades [69]; and conflict resolution, communications, enthusiasm, and overall group work attitude [70]. Students who self-select their groups spent the most amount of time outside the classroom working on course materials with their group members and felt more connected to their group members than the students in instructor-designed groups or randomly-formed groups [44]. A more recent study that assessed the impact of different group formation types on students' attitudes towards group work in large biology classrooms, found that heterogeneous groups in terms of competence have higher group work attitudes and groups formed by students were as heterogeneous as groups formed by the instructor [2].

One argument against student-selected groups is that they are often less diverse, either demographically or academically. It has been argued that more homogenous groups could negatively influence students' performance or attitudes. While students may often form more demographically homogenous groups ([29, 44], current study), these groups do not necessarily show lower outcomes [2, 44, 69–70]. We also considered the argument that people do not get to choose with whom they work in professional workspaces, and by allowing students to self-select groupmates, we are preventing our students from learning how to work with people from diverse backgrounds. While this is a valid concern and educators should help students develop skills to work with people from diverse backgrounds, we reason that making students work in particular groups without considering their preferences also does not reflect most workplace conditions where groups typically work together over longer periods of time and employees have some degree of self-selection.

These arguments may not adequately account for the potential harm of group formation methods that ignore student preferences, including the tendency for assigned groups to have negative impacts related to satisfaction, trust, and divisiveness [71]. this study suggests intentionally offering opportunities for students to build trust with others before allowing them to select their own groups. Building on this idea, we suggest that instructors' multiple goals may require multiple distinct actions. Specifically, some goals, such as promoting student satisfaction and avoiding demographic isolation can be targeted by allowing student-selected groups. Other goals, such as improving students' cultural competency skills, may require other instructional support. In practice, teaching students to work well with diverse groups of people likely requires actions beyond assigning the particular group compositions that instructors have adequate data to prescribe. While cultural competency courses can influence student attitudes in positive ways [72], we suggest there are also simple strategies instructors can use to allow student-selected groups while also providing opportunities for students to practice cultural competency skills by creating course activities that require inter-group collaboration.

The multilevel regression findings suggested that students' attitudes towards group work did not vary with being demographically isolated in lab groups. Yet, given prior research about the interpersonal challenges that isolated students sometimes face, we acknowledge that this lack of effect may not hold for all students in all contexts. Regression findings are inherently

generalizations, so they are not useful for explaining the unique experiences of individual students. Further, this study took place in a particular setting where students may have been less likely to struggle with being demographically isolated for several reasons: this sample of students was from a highly competitive institution where they were overwhelmingly high-achieving students with strong academic identities which likely minimizes both the range of academic behaviors within groups and the range of group work attitudes; student-selected groups allowed students to avoid being isolated in ways that would have felt challenging for them; and there was pedagogical support explicitly about effective group work throughout the course. Regarding the attitudinal measure, there was only a small degree of pre-post change during a course with a great deal of group work, which suggests this particular outcome measure may not be very prone to change. It is also crucial to note that these findings do not dispel concerns about other ways demographic isolation negatively influences students. Relationships between demographic isolation and other outcome measures in other contexts need to be explored as well.

## Conclusions

This study provides unique evidence about how students form groups and how their choices reduce the chances of being demographically isolated in groups. In this work, we focused on comparing the composition of student-selected groups with hypothetical randomly-formed groups. Our study shows that when given the opportunity, students mostly form groups with peers sitting near them in the classroom while a much lower proportion of students reported only considering friendship as a criterion to form groups. Groups that were formed by students led to less isolation of students identified as female and BHN+ compared to randomly-formed groups. While demographic isolation has been identified in previous studies as presenting a variety of challenges for students, our quantitative analyses did not show that demographic isolation was associated with variation in students' group work attitudes. Although instructors can implement group formation systems to create groups with particular characteristics, students may benefit more from self-selected working groups based on identities that seem important to them with other instructional supports for practicing collaboration with diverse peers. If the student-selected group formation approach is used, as educators, we recommend doing frequent check-ins to learn about group dynamics during the semester and designing activities that encourage within and between-group interactions to enhance the engagement of students with diverse identities. Comparing our findings with other group formation methods in other settings and teasing apart the conscious and unconscious biases that students' likely hold in group selection and interactions go beyond the scope of this current study and would be a valuable next step to be explored.

## Supporting information

**S1 Table. Groupwork attitude items.**
(DOCX)

**S2 Table. Multi-level linear regression output for group work EAP pretest and posttest scores.**
(DOCX)

**S3 Table. Multi-level linear regression output for group work plausible values across five trials to check for accuracy.**
(DOCX)

**S1 File.**
(PDF)

**S2 File. Data used in group selection criteria.**
(XLSX)

**S3 File. Data used in randomized grouping analyses.**
(XLSX)

**S4 File. Data used in MLM analyses.**
(CSV)

## Acknowledgments

We would like to thank the students who agreed to participate in this study, especially for their time completing the survey and sharing their experiences. We also appreciate support from the Active Learning Initiative at Cornell University.

## Author Contributions

**Conceptualization:** Mitra Asgari, Mark A. Sarvary.

**Data curation:** Mitra Asgari, Amy E. Cardace.

**Formal analysis:** Mitra Asgari, Amy E. Cardace.

**Investigation:** Mitra Asgari, Mark A. Sarvary.

**Methodology:** Mitra Asgari, Amy E. Cardace, Mark A. Sarvary.

**Project administration:** Mitra Asgari, Mark A. Sarvary.

**Visualization:** Mitra Asgari, Amy E. Cardace.

**Writing – original draft:** Mitra Asgari, Amy E. Cardace.

**Writing – review & editing:** Mitra Asgari, Amy E. Cardace, Mark A. Sarvary.

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
