## [Decision Letter · Decision Letter 0]

18 Jul 2024

PONE-D-24-13332Demographic isolation and attitudes toward group work in student-selected lab groupsPLOS ONE

Dear Dr. Asgari,

Thank you for submitting your manuscript to PLOS ONE. After careful consideration, we feel that it has merit but does not fully meet PLOS ONE’s publication criteria as it currently stands. Therefore, we invite you to submit a revised version of the manuscript that addresses the points raised during the review process.

We look forward to receiving your revised manuscript.

Kind regards,

Gabriel Velez, Ph.D.

Academic Editor

PLOS ONE

Journal Requirements:

**Additional Editor Comments:**

In general, this is a strong paper that I believe will make a contribution to the literature and is relevant for this journal. My main concerns are very much in line with those of the reviewers, and I believe revolve around considering some important qualifiers and possible limitations. In particular, the reviewers raise a number of points around possible alternative interpretations of findings (such as hidden identities, the group formation and influence, and unconscious bias).

I think the deeper points involve self-reporting. While I recognize it is often necessary, I think it is important to qualify and reshapes what can be asked from data. I very much believe these concerns can be addressed, but also consider them to be consequential enough to merit a major revision.

Reviewers' comments:

Reviewer's Responses to Questions

**Comments to the Author**

1. Is the manuscript technically sound, and do the data support the conclusions?

Reviewer #1: Partly

Reviewer #2: Partly

2. Has the statistical analysis been performed appropriately and rigorously? 

Reviewer #1: Yes

Reviewer #2: Yes

3. Have the authors made all data underlying the findings in their manuscript fully available?

Reviewer #1: No

Reviewer #2: Yes

4. Is the manuscript presented in an intelligible fashion and written in standard English?

Reviewer #1: Yes

Reviewer #2: Yes

5. Review Comments to the Author

Reviewer #1: This study investigates how demographic isolation in student-selected lab groups affects students' attitudes toward group work. Using a pre-post survey design, the authors collected data from a large-enrollment biology laboratory course to examine the criteria students use to form lab groups, the prevalence of demographic isolation in student-selected versus randomly-formed groups, and the association between demographic isolation and students' group work attitudes. The study found that students primarily formed groups with peers sitting near them, resulting in less demographic isolation than randomly-formed groups. The regression analyses showed no consistent negative associations between demographic isolation and students' attitudes toward group work.

This study is relevant given that group work is a commonly used student-centred pedagogical approach in college classrooms. Given concerns about the social-emotional challenges some students face when isolated by particular identities, educators can benefit from a better understanding of student behaviours and related social experiences in these group work settings.

However, I have some comments and questions that could potentially improve the overall quality of the study:

- Clarification on survey options: In the post-course survey, students were given six choices for criteria they considered when forming their lab groups: “friends,” “students sitting next to me in the lab,” “students’ GPA and familiarity with the biological concepts,” “students’ gender identity,” “students’ race/ethnicity,” and “students’ year in college.” Could you please clarify how you determined these specific options for the survey? Were they based on previous literature, preliminary research, or another rationale?

- Potential bias in group formation criteria: Regarding the criteria provided for group formation, don't you think limiting the possible choices to a relatively small number may bias the study's results? Additionally, do you consider self-reporting problematic, given the bias that political correctness may introduce? How did you address these potential biases in your research?

- Sample size clarification: When you mention, "The responses of students who selected non-binary or prefer not to say options had to be eliminated from the quantitative analysis due to their small sample size," how small was this sample size? While the answer is provided in Table 1, adding this information directly to the text would make the paragraph more straightforward. Could you include the specific numbers in the text for clarity?

- Unconscious bias in seating choices: The authors state, "In this examination of student survey data, we found that the majority of students reported casually forming groups with peers who were sitting around them and that their process yielded groups with less demographic isolation than would have resulted from randomly-formed groupings." Could it be that students choose their seats based on unconscious bias towards minorities? How did your analysis account for potential unconscious biases in seating choices?

Reviewer #2: Accept with minor revisions

This study aimed to address the following 3 questions:

Q1. What criteria do students consider when forming lab groups? Q2. How does the composition of student-selected groups differ in terms of demographic isolation when compared to randomly-formed groups? Q3. Do students’ attitudes toward group work vary between students who were demographically isolated in groups and those who were not?

Questions 2 and 3 were adequately address in the data, results section, and discussion. The finding that student-selected groups decreased demographic isolation is novel and provides instructors with both important practical information information to consider when assigning group work. Especially, as the paper stated, group work is one of the most common student centered practices in the college classroom. This finding, coupled with the positive perception of group work from the students provides instructors with more information to consider when attempting to create an comfortable and inclusive classroom and researchers foundational data to build upon when looking at student centered practices that are predicated on formal group work.

While question 1 was addressed by the research team, there are still unaddressed questions. The fact that students working in informal groups for 3 week prior to creating teams was not addressed as a potential impact on team selection. Figure 2 shows that most students opted to work with students near them, but the fact that those students had potentially worked together and formed trust over the course of those 3 weeks is not addressed. In a follow up study, it would be interesting to look at the group of students that did not team up with those around them, presumably those students that they’ve been working with, and collect qualitative data on their reasoning for picking a team with students they did not work with during the initial 3 weeks of the lab. This could potentially help address the claim on page 21 “students can choose groups to avoid the types of demographic isolation that they think will likely cause them harm.”. While that data does show that allowing students to choose their groups can allow them to actively avoid demographic isolation, which is interpreted as a self protective mechanism, and is important in supporting historically marginalized groups in STEM, the data does not support that there is the same impact for hidden identities (claim made on page 20). The two potentially hidden identities surveyed for are first generation status and international status. These two identities may not be immediately visible from someones physical appearance and showed no statistically significant difference in demographic isolation percentage between the classroom and simulated data. More information is needed for the authors to properly make this claim.

The suggestions for instructors in the discussion section felt thorough and nuanced. They addressed the different arguments both for and against allowing students to choose their own groups. Specifically, the argument around cultural competency was salient. The following below was a great addition to the ongoing conversation on this topic and gave practical considerations for instructors when it comes to harm mitigation and the extra emotional labor that marginalized students take on in group interactions.

"Other goals, such as improving students’ cultural competency skills, may require other instructional support. In practice, teaching students to work well with diverse groups of people likely requires actions beyond assigning the particular group compositions that instructors have adequate data to prescribe. While cultural competency courses can influence student attitudes in positive ways (Patterson et al., 2018), we suggest there are also simple strategies instructors can use to allow student-selected groups while also providing opportunities for students to practice cultural competency skills by creating course activities that require inter-group collaboration.

Overall, this is a well-done paper and resulted in an interesting observation on the topic of demographic isolation in group work.

6. PLOS authors have the option to publish the peer review history of their article (what does this mean?). If published, this will include your full peer review and any attached files.

Reviewer #1: No

Reviewer #2: No

---

## [Author Response · Author response to Decision Letter 0]

24 Aug 2024

Rebuttal letter

Re: manuscript PONE-D-24-13332

August 15, 2024

To: The Academic Editor PLOS ONE

Dear Dr. Velez,

Firstly, the authors would like to express their gratitude for the constructive feedback and suggestions received from PLOS ONE reviewers and the academic editor. We are delighted that PLOS ONE appreciates the value of our study and is interested in publishing this work upon addressing the received comments and suggestions. Below, we have responded to and addressed the main suggestions and concerns raised by the academic editor and reviewers. The related changes have been made and marked within the text as instructed.

With regards to the journal requirements item 1-3, we have made the related changes to comply with items 1 (style requirement) and 3 (providing proper caption for supporting information and in-text citation). With regards to the item 2, we followed the suggestion to share the de-identified data files as supporting information documents. 

The authors appreciate the constructive feedback and concerns shared related to the self-reporting, group formation criteria, hidden identities, and unconscious biases. We have made related changes in different sections of the manuscript in this relation to bring more clarity and improve the interpretation. Please see our detailed responses to each of the reviewers’ comments and the changes made.

Reviewer #1:

- Clarification on survey options: In the post-course survey, students were given six choices for criteria they considered when forming their lab groups: “friends,” “students sitting next to me in the lab,” “students’ GPA and familiarity with the biological concepts,” “students’ gender identity,” “students’ race/ethnicity,” and “students’ year in college.” Could you please clarify how you determined these specific options for the survey? Were they based on previous literature, preliminary research, or another rationale?

- Potential bias in group formation criteria: Regarding the criteria provided for group formation, don't you think limiting the possible choices to a relatively small number may bias the study's results? Additionally, do you consider self-reporting problematic, given the bias that political correctness may introduce? How did you address these potential biases in your research?

Authors’ response:

In the related section of the manuscript (copied below) we used the wording “including” to indicate listing only some and not all of the response choices provided to students.

“Students could select more than one option from a list of responses including the following: “friends”; “students sitting next to me in the lab”; “students’ GPA and familiarity with the biological concepts”; “students’ gender identity”; “students’ race/ethnicity”; “students’ year in college.”

Based on the reviewer 1 related comment, to improve the clarity of this section, we have included the number of answer choices provided, and shared the original question, with all answer choices, in the supporting information document (S1_file) and did in-text citation within the manuscript as well.

In terms of the logic behind the answer choices selected, in the early stage of developing the survey the authors explored a collection of STEM education research publications and brainstormed the list of most common visible and hidden demographic and academic identities students may identify and associate with. We considered the criteria that could be visible from the first day of the class when students meet each other like “racial/ethnic identity” and the ones that may be less evident or hidden but students could possibly share with each other in the first 3 weeks of the lab such as “being an international student or not”. We admit that this list might not be exhaustive, but we have tried to include here the most common visible and hidden demographic, and academic identities and background commonly studied and discussed in STEM education research. To address the reviewer’s concern, we have added a few sentences in the related section to explain our logic behind selecting these criteria for the multiple-answer question in the manuscript.

In the post-survey before students see this multiple-answer question, they were also asked to respond to an open-ended question below:

“Which criteria did you use to form your research group?”

Our intention for including this question in the survey was to explore how students’ responses may differ when they were provided a list of criteria to choose from compared to an open-ended question. Preliminary coding of students’ responses for the open-ended question showed that the “students sitting next to me in the lab” and “friends” were also the most two common criteria students shared, while other factors comprised only a very small proportion of responses. Thus, due to similar findings in open-ended and multiple-answer questions to avoid redundancy, we focused on the multiple-choice responses. 

We have added a few sentences in the manuscript to communicate this part (page 10). 

We agree that self-reporting may have its own limitations and as the only source of data collection might not be adequate enough in some cases. However, to study a question like this one we think hearing from students about what criteria mattered to them when they formed or joined a group is a more suitable choice. While self-report has inherent limitations, it can still provide valuable information about students’ perspectives. The purpose of RQ 1 was to provide an opportunity for students to share about their experiences and we would not want to omit their perspectives. The purpose of RQ 2 was to compare self-selected to random groupings, an analysis that does help reveal biases in students’ group formation processes.

We acknowledge that current study cannot tease apart the conscious and unconscious biases that students’ likely hold when locating themselves in group settings, interacting with new people, and navigating student-selected group formations. We also note that students may not be entirely forthcoming about their considerations. Yet, there was a consistent trend in students’ responses that proximity and friendship were relevant for their choice of lab partners. To address the reviewer’s concern, we have added a few sentences in the manuscript sharing what this study could not address.

- Sample size clarification: When you mention, "The responses of students who selected non-binary or prefer not to say options had to be eliminated from the quantitative analysis due to their small sample size," how small was this sample size? While the answer is provided in Table 1, adding this information directly to the text would make the paragraph more straightforward. Could you include the specific numbers in the text for clarity?

Authors’ response:

We agree with this reviewer’s comment thus we have added the related numbers to the text (page 12).

- Unconscious bias in seating choices: The authors state, "In this examination of student survey data, we found that the majority of students reported casually forming groups with peers who were sitting around them and that their process yielded groups with less demographic isolation than would have resulted from randomly-formed groupings." Could it be that students choose their seats based on unconscious bias towards minorities? How did your analysis account for potential unconscious biases in seating choices?

Authors’ response:

We agree it is possible that the selection of where to sit in lab could have been impacted by unconscious bias. As noted above, teasing apart conscious and unconscious biases in student decisions is beyond the scope of this study. Very few students reported that they intentionally considered demographics in their group choice, although students may not be entirely forthcoming about their considerations. Given student responses, we propose three logical possibilities: it may be the case that students did not want to report that they did consider demographics; it may also be the case that demographics influenced group choice only subconsciously; it may be the case that demographics did not generally influence students’ group formation processes. While teasing apart the first two possibilities is outside the scope of this study, our analyses comparing actual to simulated random groupings do address the third.

To address this comment and add clarity to this discussion we added some explanations to different parts of the manuscript. 

Reviewer #2:

-While question 1 was addressed by the research team, there are still unaddressed questions. The fact that students working in informal groups for 3 week prior to creating teams was not addressed as a potential impact on team selection. Figure 2 shows that most students opted to work with students near them, but the fact that those students had potentially worked together and formed trust over the course of those 3 weeks is not addressed. In a follow up study, it would be interesting to look at the group of students that did not team up with those around them, presumably those students that they’ve been working with, and collect qualitative data on their reasoning for picking a team with students they did not work with during the initial 3 weeks of the lab. This could potentially help address the claim on page 21 “students can choose groups to avoid the types of demographic isolation that they think will likely cause them harm.”. While that data does show that allowing students to choose their groups can allow them to actively avoid demographic isolation, which is interpreted as a self protective mechanism, and is important in supporting historically marginalized groups in STEM, the data does not support that there is the same impact for hidden identities (claim made on page 20). 

Authors’ response:

It is true that students had three weeks to get to know each other before forming groups and that could have impacted group formation. The main point of providing three weeks of interaction before forming groups, in our context, was to help students get to know each other better and ease in to working in a longer-term group. We tried to assess this by asking students a few questions in the second semester, fall 2019, post survey about how they chose where to seat in the lab and if they changed their seat between week 1 and the week they formed groups, week 4. As it was shared in page 16, students stated that on the first day of class, they chose the first available lab seat (74%) or sat by their friends (21%) in response to the questions on how they decided where to sit in the lab. The majority of students also shared not changing their seat between the first week of class and when forming their groups in week four (87%). This could indicate that the student’s initial evaluation of their peers on the first day of class based on what matters to them, could have impacted where to sit and later form groups. However, we admit that we have not assessed here the relationship between selection of where to sit in the lab and the possible impacts of trust building within the first three weeks on group formation in this study. A future study that compares students’ choices in group formation on week one vs. for example week three, when they had time to build trust, would be valuable.

To add clarity to this discussion considering what we have actually assessed, based on this comment, we have added more explanation to different parts of the manuscript, please see page 22-23.

-The two potentially hidden identities surveyed for are first generation status and international status. These two identities may not be immediately visible from someones physical appearance and showed no statistically significant difference in demographic isolation percentage between the classroom and simulated data. More information is needed for the authors to properly make this claim.

Authors’ response:

We agree that the first-generation status and international status are not as visible and easy to detect as the two other demographics. However, we think in the first three weeks of labs when students had the chance to interact with each other, and hopefully develop trust, they might have noticed or share the identities that might be hidden in their groups and seem important to them before forming groups. 

As we discussed in page 20 (now page 22-23), no statistical significance in demographic isolation percentage between the classroom and simulated data could in part be explained as “Because the proportion of students who identified as first-generation or international was low in the course, there was more likelihood for those students to be isolated in lab groups, whether they self-selected or were randomly assigned.” 

To add clarity to this discussion considering what we have actually assessed, based on this comment, we have added more explanation to different parts of the manuscript, please see pages 22-23.

We hope the academic editor finds our responses and related changes with regards to the reviewers’ comments satisfying and considers our manuscript, for publication in PLOS ONE. 

Thank you for your consideration and looking forward to hearing from you!

Sincerely,

Mitra Asgari, Ph.D. (corresponding author)

Assistant Teaching Professor

Division of Biological Sciences, University of Missouri-Columbia

Columbia, Missouri

Email: mitra.asgari@missouri.edu

---

## [Decision Letter · Decision Letter 1]

9 Sep 2024

Demographic isolation and attitudes toward group work in student-selected lab groups

PONE-D-24-13332R1

Dear Dr. Asgari,

We’re pleased to inform you that your manuscript has been judged scientifically suitable for publication and will be formally accepted for publication once it meets all outstanding technical requirements.

Kind regards,

Gabriel Velez, Ph.D.

Academic Editor

PLOS ONE

Additional Editor Comments (optional):

Reviewers' comments:

Reviewer's Responses to Questions

**Comments to the Author**

1. If the authors have adequately addressed your comments raised in a previous round of review and you feel that this manuscript is now acceptable for publication, you may indicate that here to bypass the “Comments to the Author” section, enter your conflict of interest statement in the “Confidential to Editor” section, and submit your "Accept" recommendation.

Reviewer #1: All comments have been addressed

Reviewer #2: All comments have been addressed

2. Is the manuscript technically sound, and do the data support the conclusions?

Reviewer #1: Yes

Reviewer #2: (No Response)

3. Has the statistical analysis been performed appropriately and rigorously? 

Reviewer #1: Yes

Reviewer #2: (No Response)

4. Have the authors made all data underlying the findings in their manuscript fully available?

Reviewer #1: Yes

Reviewer #2: (No Response)

5. Is the manuscript presented in an intelligible fashion and written in standard English?

Reviewer #1: Yes

Reviewer #2: (No Response)

6. Review Comments to the Author

Reviewer #1: (No Response)

Reviewer #2: N/A

7. PLOS authors have the option to publish the peer review history of their article (what does this mean?). If published, this will include your full peer review and any attached files.

Reviewer #1: No

Reviewer #2: No

---

## [Editor Report · Acceptance letter]

16 Sep 2024

PONE-D-24-13332R1 

PLOS ONE

Dear Dr. Asgari, 

I'm pleased to inform you that your manuscript has been deemed suitable for publication in PLOS ONE. Congratulations! Your manuscript is now being handed over to our production team.

Kind regards, 

on behalf of

Dr. Gabriel Velez 

Academic Editor

PLOS ONE